# Characterization of Feline Basophils on the Sysmex XN-1000V and Evaluation of a New WDF Gating Profile

**DOI:** 10.3390/ani14233362

**Published:** 2024-11-22

**Authors:** Javier Martínez-Caro, Josep Pastor

**Affiliations:** Servei d’Hematologia Clínica Veterinària, Departament de Medicina i Cirurgia Animals, Facultat de Veterinària, Universitat Autònoma de Barcelona, 08193 Bellaterra, Spain; javier.martinez.caro@uab.cat

**Keywords:** basophil, cat, eosinophil, hematology analyzer, granulocyte, method comparison

## Abstract

The purpose of this study was to evaluate a new gating method on the Sysmex XN-1000V WDF channel to improve basophil detection and quantification. We analyzed scattergrams from feline cases with basophilia, created a new gate on the WDF channel, and applied it to 9 identified basophilia cases and 34 additional cases. The comparison study showed that the new method of basophil quantification using the WDF scattergram correlated better with the manual method than the Sysmex XN-1000V method using the WNR scattergram The new WDF gating method provides more accurate basophil quantification, improving their detection in feline blood samples.

## 1. Introduction

Basophils, similar to mast cells, participate in T-helper lymphocyte Type 2 mediated immunity and Type 1 hypersensitivity reactions. Interleukin 3 serves as the primary growth factor for basophils, facilitating their production, differentiation, and survival [1]. Basophilia is not common in mammals. It is usually associated with an immunoglobulin E-mediated disorder, and frequently accompanied by eosinophilia. Main considerations should include allergic and parasitic diseases, including ectoparasites as well as gastrointestinal and vascular parasites [1]. Other conditions that may be associated with basophilia include neoplasia and other nonallergic inflammatory conditions [1], such as lymphocytic-plasmacytic and eosinophilic enterocolitis [2]. Paraneoplastic basophilia has been described as secondary to alimentary T-cell lymphoma [3,4,5,6], chronic myeloid leukemia [7], suspected acute leukemia of unidentified lineage [8], acute myeloid leukemia with basophilic differentiation [9], and acute monoblastic leukemia with chronic basophilic leukemia [10].

In veterinary medicine, basophil identification is problematic with most hematology analyzers, and feline basophils are not detected by the ADVIA 2120 [11], Sysmex XT-2000*i*V [12], Sysmex XN-V [13], or CELL-DYN 3500 [12]. This is likely because canine and feline basophils, in comparison with human basophils, are not resistant to lysis by strong surfactant, which is the method used to identify basophils on the ADVIA and Sysmex systems [11].

The aim of this study was to identify and characterize feline basophil distribution on the WDF channel from the Sysmex XN-1000V, and to explore the performance of a new basophil gate on the WDF channel in this species.

## 2. Materials and Methods

### 2.1. Preliminary Phase

We retrospectively identified cases of feline complete blood counts (CBCs) performed in our laboratory using the Sysmex XN-1000V (Sysmex Corporation, Norderstedt, Germany), along with a peripheral blood smear review, from January 2022 to May 2024. The blood smears were stained on a Hematek 2000 (Siemens Healthcare Diagnostics, Deerfield, IL, USA) using a modified Wright stain (Epredia-Richard-Allan Scientific modified Wright stain pack, Thermo Fisher Scientific, Waltham, MA, USA). The reported white blood cell (WBC) differential was based on a manually calculated differential from a 100-cell count (when judged clinically necessary it was calculated from a 200-cell count). The cases reporting basophilia (classified as >100 basophils per microliter) were identified. The basic demographic information and clinical diagnosis were summarized, and the WDF and WNR channel scattergrams were reviewed.

A new gate on the WDF channel was created using the manual analysis (extended) in software version 3 (3.07–00, Sysmex) to include the suspected basophil region. The gate information for the WDF channel in the manual analysis (extended) is shown in Appendix A. We prospectively recruited 34 new randomly selected cases of feline CBC performed in the Sysmex XN-1000V with a manual WBC differential based on a 200-cell count. The new WDF gate strategy was applied retrospectively to the basophilia cases and prospectively to the newly selected cases. A visual inspection of the new WDF scattergrams was conducted, and when appropriate, manual adjustment was performed.

Three methods for basophil quantification were considered in this work. (A) The manual method involved calculating the percentage of basophils from the WBC differential in a peripheral blood smear review (based on a 200-cell count, and 100- or 200-cell counts in the prospective and retrospective cases, respectively). (B) The WNR method involved reporting the basophil percentage from the basophil gate on the WNR channel. (C) The new WDF method involved calculating the basophil percentage from the new basophil gate on the WDF channel.

The imprecision of each method was estimated through a short-term replication study (repeatability), in which a sample was analyzed five times for the WNR and WDF methods and ten times for the manual method (WBC differential from a 200-cell count in a peripheral blood smear review).

### 2.2. Comparison Study

The three methods for basophil quantification were compared in either relative (percentage of basophils) or absolute (basophils per microliter) terms. Absolute basophil concentration was determined by multiplying the basophil percentage by the WBC concentration, the latter determined in the WNR channel.

A comparison of methods strategy, as proposed in the American Society for Veterinary Clinical Pathology (ASVCP) guidelines, was followed [14]. Normality was assessed using the Shapiro–Wilk test. For each comparison, the correlation coefficient (r), Passing–Bablok regression, and a difference plot (Bland–Altman) were calculated. Data collection was performed on Microsoft Excel software (16.78.3 version), and statistical analysis was conducted in GraphPad Prism 9 software, except for the Passing–Bablok regression fit, which was carried out on the website created by Bahar et al. (2017): https://bahar.shinyapps.io/method_compare/ (accessed on 29 June 2024) [15].

## 3. Results

### 3.1. Preliminary Phase

From January 2022 to May 2024, we identified a total of 876 CBCs from cats performed on the Sysmex with a peripheral blood smear review. A basophilia was diagnosed in nine cases (1%). The mean absolute count was 203 basophils per microliter (range from 126 to 380). The mean age of cats with basophilia was 6 years and 9 months, with a range of 10 months to 17 years. The most common breed was the European Shorthair (6/9), and one each of Persian, British, and Siamese cats. Three cats were neutered males, three cats were intact females and three were neutered females.

Two cats had chronic kidney disease, one of which was being treated with darbepoetin for its anemia of kidney disease. Two cats had been diagnosed with feline infectious peritonitis, one of which had been treated with GS-441524 (Mutian Xraphconn). One cat was polytraumatized. Another cat was treated with doxycycline and prednisolone for anemia secondary to Mycoplasma (*Candidatus* Mycoplasma turicensis). Another cat was being monitored after an enterectomy for a foreign body, and another for hyperthyroidism. Finally, one cat presented with nasolacrimal conduct bleeding, but no investigation or definitive diagnosis was made.

The WNR scattergrams of these animals were reviewed (Appendix A), but no consistent pattern was identified. In some animals, a very small number of scattered yellow dots corresponding to the basophil region were observed. On the WDF scattergrams (Figure 1), a repetitive pattern was identified in seven out of nine animals (Figure 1B–G,J). This pattern consisted of a small dot subpopulation on the eosinophil region (red dots), located above the main population, with a higher *y*-value (side fluorescence light) and a similar *x*-value (side scatter light). This smaller, characteristic, eosinophil subpopulation was thought to actually be basophils.

A new gate on the WDF channel was created through manual analysis (extended) to include the suspected basophil region (Figure 2). It was intended to best fit the higher number of cases with basophilia. The new gating setting was saved as a new species on the analyzer to allow it to be applied to other cases (prospectively and retrospectively). The new WDF setting was applied retrospectively to the 9 basophilia cases, and prospectively to the 34 new randomly selected cases (Appendix A). Nine cases (two in the basophilia group and seven in the prospective group) required manual adjustment due to significant non-basophil interference within the basophil gate (usually from the neutrophil region), significant misclassification of cell populations (gate lines drawn through the middle of a cell cluster), or inclusion of debris in the neutrophil and/or lymphocyte gates. During manual adjustment, the separation between eosinophil and basophil gates did not require modification in any of the cases.

The within-run imprecision was estimated for the three methods of relative basophil quantification and is shown in Table 1.

### 3.2. Comparison Study

The data obtained with each method for basophil quantification were not normally distributed, and nonparametric tests were used. The results of the comparison of the three methods are shown in Table 2. The Passing–Bablok regression scatterplots are shown in Figure 3, and the difference plots (Bland–Altman) are shown in Figure 4. To summarize, in the method comparison study, the strongest correlation was found between the manual and the new WDF method for both relative (%) and absolute (cells/microliter) basophil quantification. According to the Passing–Bablok regression, the smallest intercept was found for the manual versus WNR method comparison, indicating the smallest systematic error. In the manual versus new WDF method comparison, the intercept was slightly higher, indicating a small but noticeable systematic difference. The slope was closer to one for the manual versus new WDF method comparison, indicating that the new WDF method had the least proportional error and showed a more proportional relationship with the manual method. The new WDF method shows less bias and narrower limits of agreement with the manual method compared to the WNR method.

## 4. Discussion

The present work describes the precise location of feline basophils on the Sysmex XN-1000V WDF scattergram. Our findings indicate that the new WDF method for basophil quantification offers improved alignment with the manual method, with minimal bias and narrower limits of agreement compared to the WNR method.

Basophil quantification is problematic in human and veterinary medicine. Amundsen and collaborators (2012) [16] evaluated the performance of three automated hematology analyzers (Sysmex XE-2100, ADVIA 120, and CELL-DYN Sapphire) with human peripheral blood and concluded that the basophil count of the three instruments was of unsatisfactory quality when compared to flow cytometry. In veterinary medicine, basophil identification is troubling with most hematology analyzers. The feline basophils are not detected by the ADVIA 2120 [11], Sysmex XT-2000*i*V [12], Sysmex XN-V [13], or CELL-DYN 3500 [12]. The veterinary software of Sysmex and ADVIA have chosen to report basophils based on the principle that human basophils are less prone to perforation and retain their integrity in a strong surfactant while other WBCs are lysed [11]. This cell evaluation is performed on the WNR channel (Sysmex XN-V), WBC/BASO channel (Sysmex XT-*i*V), and BASO channel (ADVIA). Few veterinary species, such as rabbits, are known to have lysis-resistant basophils and are therefore suitable for reporting true basophil counts using the Sysmex XN-V [17] and ADVIA 2120 [11]. Consistent with the literature, the cases of basophilia in our work did not contain a significant number of events in the basophil region of the WNR scattergrams. The ADVIA and Sysmex analyzers may report false positive basophil identification if the samples contain pathologic cells that are lysis-resistant [16]. Other veterinary hematology analyzers, such as the ProCyte Dx or the scil VCell 5, have chosen to report basophil counts calculated from the differential WBC channel, but frequently report incorrect basophil counts in dogs and cats [18,19] and confirmation of basophilia with peripheral blood smear review is mandatory.

The WBC analysis on the Sysmex XN-V uses flow cytometry where a semiconductor laser beam (wavelength 633 nm) is directed at the cells, and measurements regarding forward scatter light signal, side scatter light signal, and side fluorescence light signal are recorded. The WNR channel counts WBCs, nucleated RBCs, and basophils, while the WDF channel classifies WBC populations in lymphocytes, monocytes, neutrophils, and eosinophils. These channels use similar reagent components, such as organic quaternary ammonium salts, non-ionic surfactant polymethine dye, and ethylene glycol, but at different concentrations [20].

The prevalence of feline basophilia in our population is 1%, significantly lower than other observations reporting 8–10% [5,19]. Possible reasons include some possible degree of underestimation of basophils during manual WBC differential counts at our institution, or population-related factors such as genetic differences, geographic influences, differences in disease incidence, or other currently unidentified factors. Due to the small number of cases with basophilia and the fact that the clinical diagnosis of these cases included several common feline diseases, a clear relationship between any clinical condition and basophilia was precluded. One case was treated with darbepoetin, but to the author’s knowledge, no clear effect of this compound on basophil or other granulocyte production has been described. Another possibility is that some degree of hypersensitivity reaction may have developed, as some erythropoiesis-stimulating agent formulations can induce an immunoglobulin E-mediated response with basophil activation, directed at the specific agents or excipients [21,22].

The scattergram regions previously described for feline basophils in the Sysmex XT-2000*i*V DIFF channel [23] and the ProCyte Dx WBC Run channel [18,24], and canine basophils in the Sysmex XT-2000*i*V DIFF channel [11] and the Sysmex XN-V WDF channel [25], are lower on the *y*-axis and *x*-axis than the proposed basophil gate in the WDF channel on the Sysmex XN-1000V that we propose in the present work. Recently, Guerlin and colleagues [26], and Mato-Martín and colleagues [27], described a similar location of feline basophils on the WDF scattergram, further confirming our hypothesis and findings. Furthermore, applying a manual gating strategy similar to ours on the WDF scattergram yielded basophil counts comparable to those obtained by manual counting [27].

Those previous regions [11,18,23,24,25] are closer to the neutrophil gate than ours and could easily be affected by toxic change and/or left shift in neutrophils in dogs, or by platelet clumps in cats. The proposed new basophil gate is located higher than the eosinophils on the *y*-axis and in a region where no other WBCs are typically located, and from our perspective, it may be less affected by pathologic WBCs than the described basophil area of feline and canine basophils in other analyzers, making it more suitable for use in daily practice. However, no gating strategy is free of cell misclassification. In our caseload, we performed manual adjustment of the gates in a few cases (9 out of 43), mainly due to significant non-basophil interference within the basophil gate (frequently from the neutrophil region), but also due to significant misclassification of other cell populations, or due to inclusion of debris in the neutrophil and/or lymphocyte gates. It is important to mention that the separation between eosinophil and basophil gates was not modified and remained constant throughout the cases.

The differences between the proposed new basophil gate on the Sysmex XN-1000V and the basophil region previously described for cats on the Sysmex XT-2000*i*V [23] may be explained by technical differences between the analyzers. The WBC differential of the Sysmex XN-V series is performed on a new channel, the WDF channel, which is comparable with the previous DIFF channel of the Sysmex XE and XT analyzers. The scattergrams of the two channels display distinct patterns, which are attributed to variations in the reagents used, as well as differences in the hardware and software [28]. Kawauchi and colleagues (2014) [28] evaluated the staining and physical effects of WDF and DIFF reagents on the human WBC. They assessed the fluorescence staining intensity by confocal laser scanning microscopy, the intracellular structure by transmission electron microscopy, and the size and surface structures by scanning electron microscopy. Fluorescence staining with the WDF reagent was found mainly in the cytoplasm, and it was thought that the polymethine dye stained the nucleic acids in the cytoplasm. The staining intensity was higher in monocytes, T lymphocytes, B lymphocytes, neutrophils, and eosinophils, in that order, and correlated with the fluorescence intensity on the WDF scattergram. Basophils were not evaluated in that study, but evaluation of feline basophils by confocal laser scanning microscopy after treatment with WDF reagents could help to verify the position on the WDF scattergram. They also found that the WDF reagent caused less damage to the cell membrane than the DIFF reagent, and the intracellular structure of the WBCs was better preserved after treatment with WDF reagents. This explained why monocytes on the Sysmex XN were better separated from lymphocytes [28]. Similarly, we hypothesize that this may be the reason why basophils have similar side scatter values to eosinophils on the Sysmex XN-V, rather than lower side scatter values as reported on the Sysmex XT-*i*V [23].

Using light microscopy and Romanowsky staining, basophils are very characteristic cells, with a highly species-dependent cytomorphology. Feline basophils are larger than an eosinophil, with a variably shaped nucleus (ribbonlike, bi-, tri-lobed, or U-shaped) with moderately condensed chromatin and numerous relatively uniform, round-to-oval, lavender, cytoplasmic granules [5]. In general, basophil granule size is slightly larger than that of eosinophil granules (personal observation, Figure 5). Unlike other species in which basophil characteristics are profoundly dependent on the staining method, the secondary granules of feline basophils stain pale lavender (without metachromasia) with both the aqueous and methanolic Romanowsky methods [29].

The H20-A2 document from the Clinical and Laboratory Standards Institute (CLSI) states that the reference method for basophil identification is by microscopic slide examination [16]. However, the CLSI also states that when cells are less frequent than 5%, manual counting is not suitable as a reference method because of its high imprecision [16]. Other studies have used flow cytometry as a reference method for basophil quantification [16], but to our knowledge, the identification of feline basophils by flow cytometry has not been validated to date. Given these arguments, we have considered the manual method to be the only one that could confirm the presence of basophils and the closest to a gold standard for comparison with the WNR and WDF methods.

In the short-term (within-run) imprecision study, the new WDF method demonstrated significantly superior precision compared to both the manual and WNR methods, with the latter showing substantial imprecision. In the method comparison study, the new WDF method showed better correlation, better agreement, and less bias with the manual method for both % basophils and absolute basophils per microliter compared to the WNR method. The statistical analyses suggest that the new WDF method is more reliable and accurate than the WNR method when compared to the manual method. The authors speculate that better correlation coefficient values may have been obtained for the comparison between the new WDF method and the manual method if cases with a higher percentage of basophils (marked basophilia) had been included in the caseload.

The ASVCP guideline for allowable Total Error in hematology [30] does not provide a recommendation for basophils because they are typically not accurately enumerated by common reference laboratory instruments. In human medicine, the Total Error allowed recommendations for basophils are 26.8% and 17.9%, minimum and desirable, respectively; however, these recommendations are based on human biological variation rather than clinical decision limits (European Federation of Clinical Chemistry and Laboratory Medicine [EFLM]—consulted https://biologicalvariation.eu/meta_calculations (accessed on 1 July 2024)) [31]. In any case, the extremely low number of basophils in peripheral blood makes it difficult to fully validate the analytical performance of any basophil detection method.

The main limitation of the present study is the mild degree of basophilia and the absence of cases with marked basophilia. Additionally, there is a lack of a true gold standard for optimal method comparison. The manual method for basophil quantification suffers from high imprecision, which may have influenced the results and negatively affected the method comparison results. Finally, the number of cases included in the present work is relatively small, and further studies with larger case numbers, especially those involving cases of marked basophilia, are recommended.

## 5. Conclusions

To the author’s knowledge, this is the first validation of feline basophil quantification using the WDF scattergram of the Sysmex XN-1000V. The presence of an additional cell population in the Sysmex XN-1000V WDF channel scattergram, as described in this paper, should alert the operators to evaluate the blood smear for possible basophilia. Basophil concentration in feline peripheral blood can be determined using a new gate on the WDF channel of the Sysmex XN-1000V, which provides better performance than the WNR channel and is comparable to the manual method. Further studies with more individuals and cases with marked basophilia are needed in order to verify the usefulness of the new basophil gate on the WDF channel in this species.

## Figures and Tables

**Figure 1 animals-14-03362-f001:**
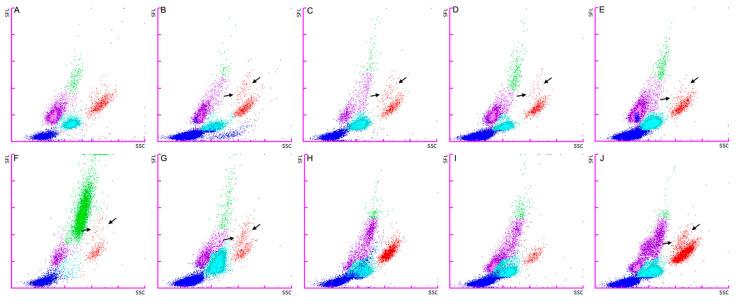
WDF scattergram for the Sysmex XN-1000V analyzer for feline blood specimen. Original, non-adjusted scattergrams. (**A**) shows an example of a healthy cat. (**B**–**J**) show feline cases with basophilia based on the blood smear basophil determination. The arrows indicate the suspected basophil population. (**F**) shows significant misclassification of toxic neutrophils counted as monocytes (personal observation, confirmed by blood smear examination). Abbreviations: SFL, side fluorescence light; SSC, side scatter light. Particle representation: clear blue dots (neutrophils), purple dots (lymphocytes), green dots (monocytes), red dots (eosinophils), and dark blue dots (debris).

**Figure 2 animals-14-03362-f002:**
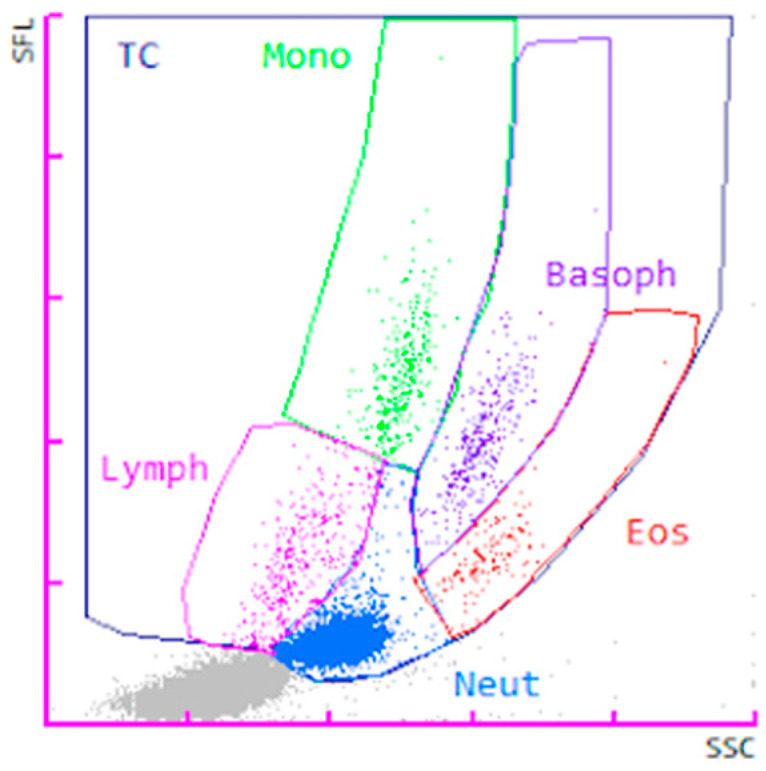
WDF scattergram for the Sysmex XN-1000V analyzer for a feline blood specimen with the new WDF gating strategy. Abbreviations: Basoph, basophils; Eos, eosinophils; Lymph, lymphocytes; Neut, neutrophils, Mono, monocytes; SFL, side fluorescence light; SSC, side scatter light; TC, total cells. Particle representation: blue dots (neutrophils), pink dots (lymphocytes), green dots (monocytes), red dots (eosinophils), purple dots (basophils), and gray dots (debris).

**Figure 3 animals-14-03362-f003:**
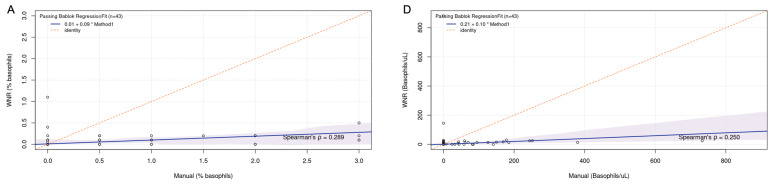
Passing–Bablok regression scatterplot of comparison of methods for basophil quantification. (**A**,**D**), comparison of manual method versus WNR method. (**B**,**E**), comparison of manual method versus new WDF method. (**C**,**F**), comparison of WNR method versus new WDF method. (**A**–**C**) methods compared using % basophils; (**D**–**F**) methods compared using absolute basophil units (basophil per microliter). The 0.95 confidence bounds are calculated with the bootstrap (BCa) method. Abbreviations: BCa, Bias-Corrected and Accelerated; uL, microliter.

**Figure 4 animals-14-03362-f004:**
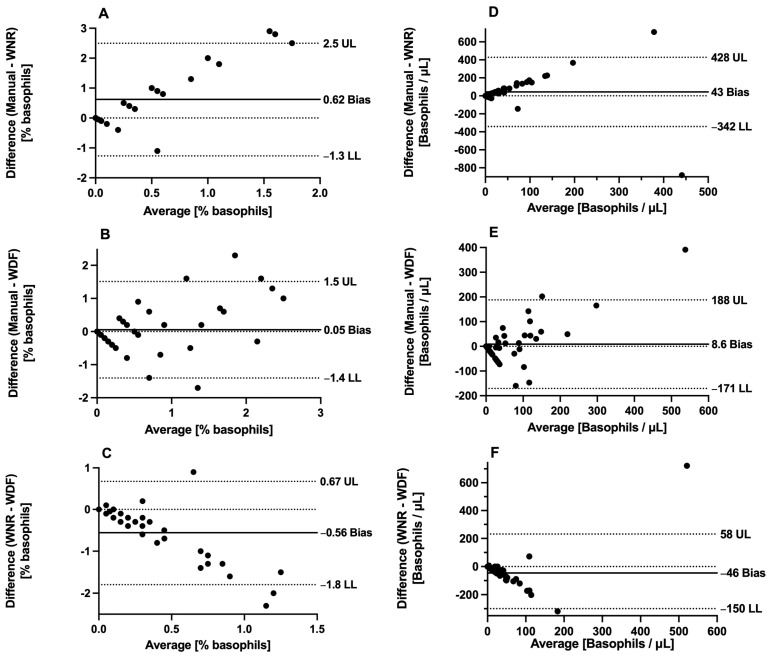
Difference versus average plot (Bland–Altman) of comparison of methods for basophil quantification. (**A**,**D**), comparison of manual method versus WNR method. (**B**,**E**), comparison of manual method versus new WDF method. (**C**,**F**), comparison of WNR method versus new WDF method. (**A**–**C**) methods compared using % basophils; (**D**–**F**) methods compared using absolute basophil units (basophil per microliter). Abbreviations: LL, lower limit of agreement; UL, upper limit of agreement; μL, microliter.

**Figure 5 animals-14-03362-f005:**
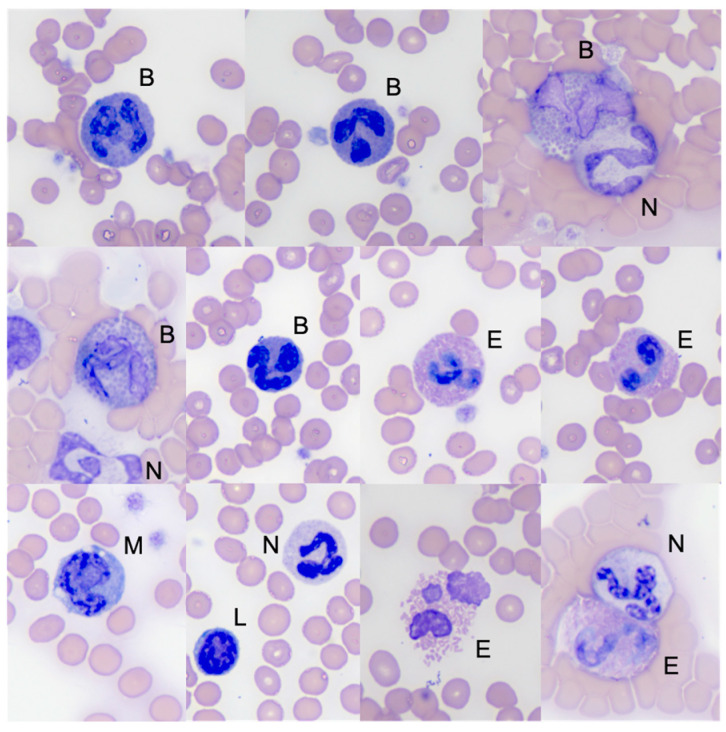
Peripheral blood smear micrographs showing feline WBCs. Note several basophils with numerous, relatively uniform, round-to-oval, lavender, secondary granules. Abbreviations: B, basophil; E, eosinophil; L, lymphocyte; M, monocyte; N, neutrophil. Modified Wright stain, original magnification ×100 objective.

**Table 1 animals-14-03362-t001:** Short-term (within-run) imprecision determinations for the three methods of basophil quantification. The basophil measurements are expressed in percentage (%) of total white blood cells.

	Manual Method	WNR Method	New WDF Method
Mean	5.9	0.02	5.3
Standard deviation	2	0.16	0.16
Coefficient of variation (%)	34	224	3.1

**Table 2 animals-14-03362-t002:** Results of the comparison of the three methods for basophil quantification. Abbreviations: CI, confidence interval; LoA, limits of agreement.

	Units	Manual Method versus WNR Method	Manual Method versus New WDF Method	WNR Method versus New WDF Method
Correlation coefficient (r)	% basophils	0.29	0.64	0.22
	Basophils/μL	0.25	0.55	0.53
95% CI of r	% basophils	−0.02 to 0.55	0.41 to 0.79	−0.09 to 0.49
	Basophils/μL	−0.06 to 0.52	0.29 to 0.74	0.26 to 0.72
*p*-value of r	% basophils	0.0603	<0.0001	0.1654
	Basophils/μL	0.1053	<0.0001	0.0003
Passing–Bablok regression intercept	% basophils	0.01	0.2	−0.6
	Basophils/μL	0.21	19	−9.23
95% CI of intercept	% basophils	0 to 0.11	0.09 to 0.24	−1.633 × 10^15^ to 1.633 × 10^15^
	Basophils/μL	0 to 19	1.66 to 21	−129.6 to 9
Passing–Bablok regression slope	% basophils	0.08	0.65	14
	Basophils/μL	0.1	0.66	8.61
95% CI of slope	% basophils	0 to 0.16	0.5 to 1.2	−1.633 × 10^15^ to 1.633 × 10^15^
	Basophils/μL	−0.06 to 0.2	0.48 to 1.56	4.45 to 19.47
Bland–Altman bias	% basophils	0.62	0.053	−0.56
	Basophils/μL	43	8.6	−34
Standard deviation of bias	% basophils	0.96	0.74	0.63
	Basophils/μL	197	91	135
95% LoA of bias	% basophils	−1.3 to 2.5	−1.4 to 1.5	−1.8 to 0.67
	Basophils/μL	−342 to 428	−171 to 188	−300 to 231

## Data Availability

All available data are presented in this manuscript.

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
