# Peer review of "Characterization of Feline Basophils on the Sysmex XN-1000V and Evaluation of a New WDF Gating Profile"

_animals, 2024, doi:10.3390/ani14233362_

Round 1

Reviewer 1 Report

Comments and Suggestions for Authors

This study demonstrates that with the Sysmex XN-1000V in cats, on the WDF scattergram, the additional and well-separated population observed above the eosinophil position is very likely to be basophils. Thus, with manual gating, a more accurate quantification of basophils can be obtained than that given automatically by the Sysmex XN-V.

Simple summary

Line 8 : delete the first sentence “Identification …. Analyzers”

Line 12-14: not clear. “The comparison study showed that the new method of basophil quantification using the WDF scattergram correlated better with the manual method than the Sysmex XN-1000V method using the WNR scattergram.”

Abstract

Line 16: change the first sentence for : “ The Sysmex XN-1000V provides a percentage and concentration of basophils from the WNR scattergram as is done for human samples, but this method has been shown to be irrelevant in cats.”

Line 18 : delate “of the Sysmex XN-1000V”

Line 25-27 : not clear. “The comparison study showed that the new method of basophil quantification using the WDF scattergram correlated better with the manual method than the Sysmex XN-1000V method using the WNR scattergram.”

Introduction

L39 : allergic and parasitic diseases

L40 : intestinal and vascular parasites.

L54: delate “preliminarily”

L55-57 : this should be delated and added in the discussion

M&M

L62-68 : it is a retrospective study and should not be detailed here but added line 73

L81-82 : To simplify, the WDF method can be call the new method and the WNR method can be called the Sysmex method and these short name can be used through the article.

L85-88: in a short-term imprecision study, the measurements must be repeated 10 times. Why were they done 3 times and 5 times?

L89 : modify the title for “comparison study”

Results

L104-5: modify the sentence for “A basophilia was diagnosed in 9 cases (1%).

L161: modify the title for “comparison study”

Figure 1 : modify the title

“WDF scattergram from the Sysmex XN-1000V analyser for feline blood specimen. …”

Figure 2 : modify the title

“WDF scattergram from the Sysmex XN-1000V analyser for a feline blood specimen with the new WDF gating strategy.. …”

Figure 3: modify the title

Passing-Bablok plots from the comparison between the new and the manual methods and the Sysmex and the manual methods for basophils quantification.

Correct in figures A, B, C, D, E and F : basophils for basophils

Delate “The 0.95 -confidence … method” on figures A, B, … and add it in the caption

Figure 4

Bland Altman plots from the comparison between the new and the manual methods and the Sysmex and the manual methods for basophils quantification.

redo the figures using manual counting as the gold standard and not the average of the two methods and modify the Table 2 accordingly.

Table 1

in a short-term imprecision study, the measurements must be repeated 10 times. Why were they done 3 times and 5 times?

Table 2

redo the figures using manual counting as the gold standard and not the average of the two methods and modify the Table 2 accordingly.

Supplementary table 1 : not clear. Delate or explain!!

Supplementary figures : no title are given.

Discussion

The discussion must be rebuilt

First paragraph : general information

L198: Answer to the question and aim of this study

Just after this sentence, add the paragraph about basophil quantification L211-L231

Second paragraph : descriptive statistics on basophilia L198-204

L205-210 : delate

Delate L232-239

Third paragragraph : identification of the basophils on WDF scattergram in cats

 L240- …

Delate L241 : “and canine … channel “

Delate (2024) L244

L245: delate “these previous …in neutrophils” for These previous area between neutrophils and eosinophils could easily be affected by platelet clumps.

L281-289 : Delate

Fourth paragraph : comparison study

L290-298 : justification that even if not good, manual method is still the gold standard.

L306: delate were and “although this was not possible”.

Delate “the higher basophile …. The manual count”.

Fifth paragraph : Limitation

Modify this paragraph. The main limitation is not the number of specimen but the very light basophilia and do case of marked basophilia.

Conclusions:

L334 : Further study with marked basophilia are needed …

L104not reviewed here.

Reviewer 2 Report

Comments and Suggestions for Authors

Dear authors,

First, congratulations on your well-written, nicely-structured and beautifully presented work. Although I have some (very) minor comments, I do not have any doubt about the merit and interest of this manuscript. Despite this topic concerning advanced applications and uses of hematology analyzer, I think that the authors are able to display their results and conclusions in a easy-to-understand way. The presentation of graphic data and the several methods for the comparison among techniques are adequate, although I would suggest to tuns some supplementary files into proper figures in order to achieve a more "round" final manuscript that gets the attention from the not-so-specialized reader (see further comment about this).

These are some minor comments that I think might improve the final manuscript.-

Line 12.- "basofilia" should be spelled basophilia.

Line 55.- Although I truly appreciate the sincerity of the authors mentioning the work by Guerlin, I do think that this statement could be more appropiate in the Discussion (when comparing your gating with previous works).

Line 67.- Please comment which author/s performed this manual differential.

Introduction.- I think it could be interesting to clarify that some previous authors (Mato-Martin et al.) have also noted that feline basophils can be gated in this analyzer in a similar region to the one here described (https://www.esvcp.org/archives/mystery-slide-selections/1013-case-4-abnormal-population-in-wdf-scattergram-sysmex-xn-v-in-a-cat-with-answers/file.html). The mentioned report is based in a single case and has not been formally published (to my knowledge). Thus, mentioning it should not be seen as an disadvantage/demerit and, instead, could strengthen your conclusions and be understood as a recognition to those previous authors. A similar mindset could apply to comparisons with the work by Guerlin.

Figure 1.- "misclassification of toxic neutrophils". I understand that this is well-known to any specialist working with this analyzer. Nonetheless, a reference could be useful for any reader not well versed (in case the authors are unable to find an adequate reference or consider it to not be appropiate, I would suggest to use the formula "personal observation").

Fig. 2.- The words "Neut" in the figure are difficult to see (and indeed, I could not see them in a black/white print).

Discussion.- One initial paragraph summarizing your major findings/results could be very useful.

Line 244.-  I would advise to firstly compare previous gating options with the one here studied and later refer to the findings made by Guerlin et al. It would be important here to clarify whether Guerlin proposed and tested a specific basoph gating or just mentioned the graph in one/various case/s of feline basophilia. Some comments comparing with the report made by Mato-Martin could also be useful.

Line 286.- I think that Supp. Fig 3 could be implemented as a proper Figure in the manuscript and could strengthen the visual impact of your work to less "specialized" reader.

Line 306.- "were have been included" is convoluted and does not sound right.

Supplementary figures.- This material is very nice and interesting, but a better description of the graphs is needed (are the letters in Suppl. Fig. 1 referring to the same cats as in Fig 1? which cases are the prospective/retrospective ones in Supp. Fig 2? Suppl. Fig. 3 would need identification for the letters used.
